# FTBNN: Rethinking Non-linearity for 1-bit CNNs and Going Beyond

## Abstract

Binary neural networks (BNNs), where both weights and activations are binarized into 1 bit, have been widely studied in recent years due to its great benefit of highly accelerated computation and substantially reduced memory footprint that appeal to the development of resource constrained devices. In contrast to previous methods tending to reduce the quantization error for training BNN structures, we argue that the binarized convolution process owns an increasing linearity towards the target of minimizing such error, which in turn hampers BNN's discriminative ability. In this paper, we re-investigate and tune proper non-linear modules to fix that contradiction, leading to a strong baseline which achieves state-of-the-art performance on the large-scale ImageNet dataset in terms of accuracy and training efficiency. To go further, we find that the proposed BNN model still has much potential to be compressed by making a better use of the efficient binary operations, without losing accuracy. In addition, the limited capacity of the BNN model can also be increased with the help of group execution. Based on these insights, we are able to improve the baseline with an additional 4∼5% top-1 accuracy gain even with less computational cost. Our code and all trained models will be made public.

## 1 Introduction

In the past decade, Deep Neural Networks (DNNs), in particular Deep Convolutional Neural Networks (DCNNs), has revolutionized computer vision and been ubiquitously applied in various computer vision tasks including image classification (Krizhevsky et al., 2012), object detection (Liu et al., 2020a) and semantic segmentation (Minaee et al., 2020). The top performing DCNNs (He et al., 2016; Huang et al., 2017) are data and energy hungry, relying on cloud centers with clusters of energy hungry processors to speed up processing, which greatly impedes their deployment in ubiquitous edge devices such as smartphones, automobiles, wearable devices and IoTs which have very limited computing resources. Therefore, in the past few years, numerous research effort has been devoted to developing DNN compression techniques to pursue a satisfactory tradeoff between computational efficiency and prediction accuracy (Deng et al., 2020).

Among various DNN compression techniques, Binary Neural Networks (BNNs), firstly appeared in the pioneering work by Hubara et al. (2016), have attracted increasing attention due to their favorable properties such as fast inference, low power consumption and memory saving. In a BNN, the weights and activations during inference are aggressively quantized into 1-bit (namely two values), which can lead to $32\times$ saving in memory footprint and up to $64\times$ speedup on CPUs (Rastegari et al., 2016). However, the main drawback of BNNs is that despite recent progress (Liu et al., 2018; Gu et al., 2019; Kim et al., 2020b), BNNs have trailed the accuracy of their full-precision counterparts. This is because the binarization inevitably causes serious information loss due to the limited representational capacity with extreme discreteness. Additionally, the discontinuity nature of the binarization operation brings difficulty to the optimization of the deep network (Alizadeh et al., 2018).

A popular direction on enhancing the predictive performance of a BNN is to make the binary operation mimic the behavior of its full-precision counterpart by reducing the quantization error cuased by the binarization function. For example, XNOR-Net (Rastegari et al., 2016) firstly introduced scaling factors for both the binary weight and activation such that the output of the

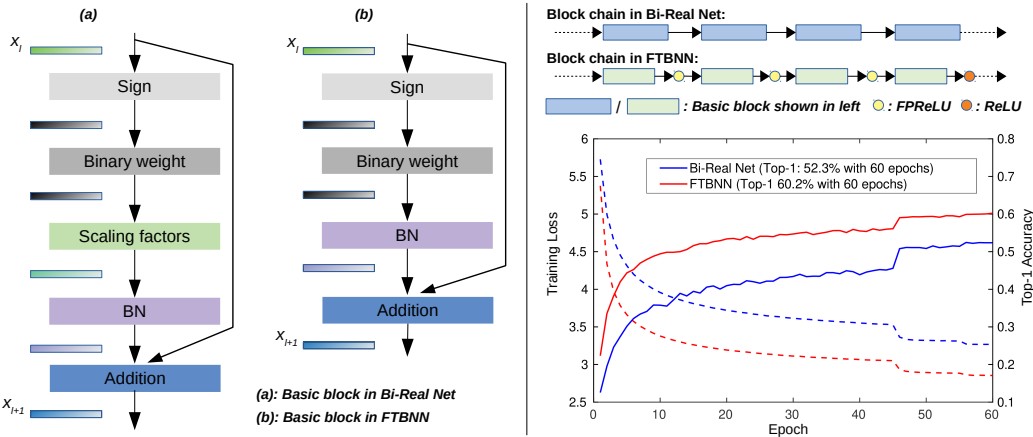

Figure 1: *Left*: The basic block in original Bi-Real Net *vs.* the simplified basic block in FTBNN, where we directly absorb the explicit scaling factors into the BN layer by leveraging BN's scaling factors. *Right*: The non-linear modules (ReLU or FPReLU) are explicitly added after each basic block in FTBNN. To maximize the model's discriminative power while keeping its training stability, the number of ReLU is controlled and the proposed FPReLU is connected with most blocks. Training curves on ImageNet of the two 18-layer networks are depicted to show the training efficiency. Solid lines denote Top-1 accuracy on the validation set (y-axis on the right), dashed lines denote training loss (y-axis on the left). Both models are trained from scratch.

binary convolution can be rescaled to closely match the result of the real-valued convolution just like the original full-precision weight and activation are used. The method outperforms its vanilla counterpart BNN (Hubara et al., 2016) by a large margin (44.2% *vs.* 27.9% in Top-1 accuracy on ImageNet using the AlexNet architecture (Krizhevsky et al., 2012)). Because of the remarkable success of XNOR-Net, a series of approaches emerged subsequently with the effort of either finding better scaling factors or proposing novel optimization strategies to further reduce the quantization error. Specifically, XNOR-Net++ (Bulat & Tzimiropoulos, 2019) improved the way of calculating the scaling factors by regarding them as model parameters which can be learnt end-to-end from the target loss. While Real-to-Bin (Martinez et al., 2020) proposed to compute the scaling factors on the fly according to individual input samples, which is more flexible. From another perspective, IR-Net (Qin et al., 2020) progressively evolved the backward function for binarization from an identity map into the original Sign function during training, which can avoid big quantization error in the early stage of training. BONN (Gu et al., 2019) added a Bayesian loss to encourage the weight kernel following a Gaussian mixture model with each Gaussian centered at each quantization value, leading to higher accuracy. Other works aiming to reduce the quantization error also include ABC-Net (Lin et al., 2017), Bi-Real Net (Liu et al., 2018), ProxyBNN (He et al., 2020), *etc.*

However, another problem arises with the quantization error optimized towards 0, especially for the structure like Bi-Real Net ( Fig. 1), where the only non-linear function is the binarization function. The non-linearity of the binarization function will be eliminated if the binary convolution with scaling factors can perfectly mimic the real-valued convolution in the extreme case (quantization error equals to 0), thus hindering the discriminative ability of BNNs. Therefore, it is necessary to re-investigate the non-linear property of BNNs when inheriting existing advanced structures.

Based on this consideration, we conduct the experiment on MNIST dataset (LeCun & Cortes, 2010) using a 2-layer Bi-Real Net like structure (which begins with an initial real-valued convolution layer and two basic blocks illustrated in Fig. 1 (b), optionally followed by a non-linear module, and ends with a fully connected (FC) layer) and some interesting phenomenons can be found as shown in Fig. 2, where we visualized the feature space before the FC layer and calculated the feature discrepancy caused by the binarization process as well as the corresponding classification accuracy (ACC, in %) for each model. Firstly, comparing the first two figures, despite the big quantization error made by binarization, the binary model achieves much higher accuracy than the real-valued model, which does not have quantization error. This indicates that the binarization function owns a potential ability to enhance the model's discriminative power, and also explains why Bi-Real Net

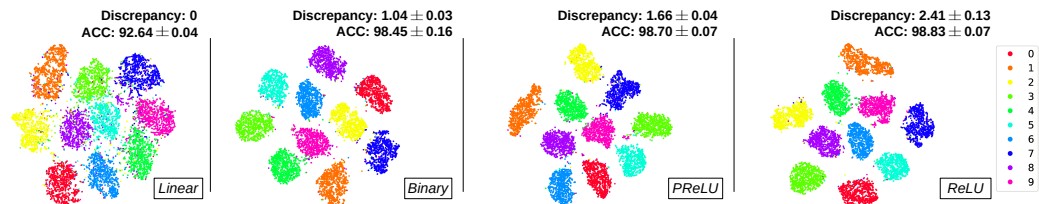

Figure 2: Feature visualization for 4 different models experimenting on MNIST, which are denoted as *Linear*: using real-valued convolution layers without non-linear module; *Binary*: using binary convolution layers without non-linear module; *PReLU*: using binary convolution layers with PReLU module; *ReLU*: using binary convolution layers with ReLU module. Testing accuracies are recorded based on 5 runs for each model. The average discrepancy between output from binary convolutions and that from its corresponding real-valued convolutions (using the proxy weights and original real-valued activations to conduct the convolution) are recorded as well (best viewed in color).

Table 1: Top-1 accuracy on ImageNet of different BNN models with commonly used training strategies, based on the ResNet-18 architecture (He et al., 2016). ✓/✗ denotes the corresponding strategy is / is not utilized. KD denotes knowledge distillation, MS means using multiple training steps, GA indicates applying gradient approximation in the backward pass. Note that BinaryDuo (Kim et al., 2020b) improved GA with a coupled ternary model. Scaling represents using explicit scaling factors to reweight the output of binary convolutions. N/A+$n$ means only the number of epochs $n$ in the final step for multi-step training is given. Here Bi-Real Net was trained using two different implementations. † indicates the double skip connections proposed by Liu et al. (2018) were used. It can be seen that FTBNN achieves promising accuracy even with a naive training scheme and can be further improved when combining the training strategies.

| Method | KD | GA | MS | Scaling | Epoch | Top-1 | Method | KD | GA | MS | Scaling | Epoch | Top-1 |
|---|---|---|---|---|---|---|---|---|---|---|---|---|---|
| BNN | ✗ | ✗ | ✗ | ✗ | N/A | 42.2% | IR-Net † | ✗ | ✓ | ✗ | ✓ | N/A | 58.1% |
| XNOR-Net | ✗ | ✗ | ✓ | ✓ | N/A+58 | 51.2% | BinaryDuo † | ✗ | ✓ | ✓ | N/A | 120+40 | 60.9% |
| XNOR-Net++ | ✗ | ✗ | ✗ | ✓ | 80 | 57.1% | Real-to-Bin baseline † | ✓ | ✗ | ✓ | ✓ | 75+75 | 60.9% |
| Bi-Real Net (B) † | ✗ | ✓ | ✗ | ✓ | 256 | 56.4% | **FTBNN** † | ✗ | ✗ | ✗ | ✗ | **60** | **60.2%** |
| Bi-Real Net (A) † | ✗ | ✓ | ✓ | ✓ | N/A+20 | 56.4% | **FTBNN** † | ✓ | ✗ | ✓ | ✗ | **75+75** | **63.2%** |

can possibly achieve high accuracy even without any non-linear modules. Such ability should be respected when we design the BNN structures rather than minimizing the quantization error to 0. Second, when explicitly introducing a non-linear module, either PReLU (He et al., 2015) or ReLU, the discriminative power can be further enhanced, even though the feature discrepancy is enlarged. This again shows that there is no strict correlation between the quantization discrepancy and the final predictive performance. Meanwhile, the ReLU model brings the most discriminative power with a clearer class separateness in the feature space. Our following experiments also show the superiority of the traditional ReLU function due to a stronger non-linearity .

Motivated by the above observation, we propose FTBNN (meaning fast training, see Fig. 1), by firstly eliminating the scaling factors after the binary convolutional operation, which are instead fused into the Batch Normalization (BN) (Ioffe & Szegedy, 2015) layers by leveraging BN's scaling factors, and explicitly configuring appropriate non-linear modules after each block. By doing so, we can release the optimization burden for extra scaling layers as well as keep the non-linear property of the Sign function via a weaker scaling mechanism. At the same time, the configuration of explicit non-linear modules (we also proposed a more flexible non-linear module, namely Fully Parametric ReLU (FPReLU) shown in Fig. 3) is able to further boost the discriminative ability of the BNN model as well as ensure the training stability. The resulted FTBNN is highly competitive among the state-of-the-art BNN frameworks, with its enhanced performance in terms of both accuracy and training efficiency, as shown in Table 1 and Fig. 1. Note that very recently Liu et al. (2020b) also demonstrated the importance of non-linearity by redistributing activations in BNNs.

To go beyond the baseline structure we have already obtained, we also notice that two aspects in the literature were rarely investigated. One is to think how far we can go to make use of the fast and light binary operations and avoid the use of expensive real-valued counterparts, to make our

BNN model more compact and computationally efficient. This is motivated by the fact that the real-valued operations in a BNN structure often take a considerable proportion of computation cost (*e.g.*, 85% in an 18-layer Bi-Real Net, and so does the proposed FTBNN, see Fig. 3). Second, inspired by Binary MobileNet (Phan et al., 2020) which leveraged group convolution (Xie et al., 2017) to automatically configure an optimal binary network architecture in their network search space, we consider how group convolution can affect the performance of BNN structures (in representational capacity, discriminative power, *etc.*) with different configurations (vary in depth and width). To the best of our knowledge this paper is also the first attempt to give such a specific investigation and can lead to useful insights on designing better BNN structures with the group mechanism.

As for our experimental outcomes, firstly, we are able to build an easily trainable strong baseline FTBNN by improving the popular Bi-Real Net structure, already achieving the state-of-the-art performance (Table 1, Table 3 and Table 4). Secondly, we enhanced the FTBNN by circumventing the usage of real-valued operations and incorporating the group mechanism, leading to a series of derived models that not only surpass the baseline in higher accuracy but also have less computational overhead (Table 4). It is also hoped that our exploration can give a better understanding of BNN design in the community and derive more efficient binary structures.

## 2 METHODS

### 2.1 BACKGROUND AND FTBNN

In order to reduce the quantization error (denote as $\mathcal{E}$) and protect the information flow in BNNs, which has been an important focus of recent years' studies, a very basic idea is introducing several scaling factors to reweight the output of the binary convolution, such that the rescaled output can approach closely to the result as if using the real-valued operation:

$$\mathcal{E} = (\mathcal{W}_b \oplus \mathcal{A}_b) \odot \Gamma - \mathcal{W} \otimes \mathcal{A} \approx 0, \qquad (1)$$

where $\mathcal{W}$, $\mathcal{A}$, $\mathcal{W}_b$ and $\mathcal{A}_b$ indicate the real-valued weight and activation, the binarized weight and activation respectively, $\otimes$ represents a multiplicative convolution and $\oplus$ represents a convolution using the *XNOR-Count* operations (or binary convolution), $\Gamma$ is the scaling factors and $\odot$ is element-wise multiplication.

In XNOR-Net (Rastegari et al., 2016), $\Gamma$ actually consisted of two separate parts, for the weights and activations respectively and both were calculated in an analytical way during the forward pass, by solving an optimization problem, which is therefore time-consuming. XNOR-Net++ (Bulat & Tzimiropoulos, 2019) further argued that $\Gamma$ can be a single part and learnt during training, showing better performance in terms of both accuracy and efficiency. More recently, Real-to-Bin (Martinez et al., 2020) proposed a data-driven version of $\Gamma$, which was computed on the fly according to individual input samples.

In fact, with the purpose of finding a satisfactory initialization for $\Gamma$, usually a pretrained model with full-precision weights is needed (Rastegari et al., 2016; Lin et al., 2017). However, it is shown that the scaling factors can be fusable with the parameters in BN (Liu et al., 2018), directly training the BN layers can avoid the above troublesome, without degenerating the whole structure (Bethge et al., 2019). This also matches our assumption on Section 1 that the BN scaling factors can be a good alternative for an explicit scaling layer.

On the other hand, to maintain the information flow and maximize BNN's expressive ability, another aspect of experience in the literature is the usage of shortcut connections (He et al., 2016). One milestone about that is the Bi-Real Net structure (Liu et al., 2018), where the authors emphasized using the shortcut which connects the real-valued activations before the Sign function to the output of binary convolution, and proved the almost computation-free shortcut can significantly preserve BNN's representational capability. It was then repeatedly adopted in other works (Liu et al., 2020b; Bulat et al., 2020; Kim et al., 2020a) to improve the quality of information propagation.

Standing on the shoulders of giants and according to our discuss in Section 1, we propose a simple block based structure, dubbed as FTBNN[1] (Fig. 1) that both enables training efficiency and encourages non-linearity by discarding the scaling factors which are implicitly fused into BN, and

---

[1]For reduction block, we adopt the real-valued 1x1 convolutional downsampling shortcut.

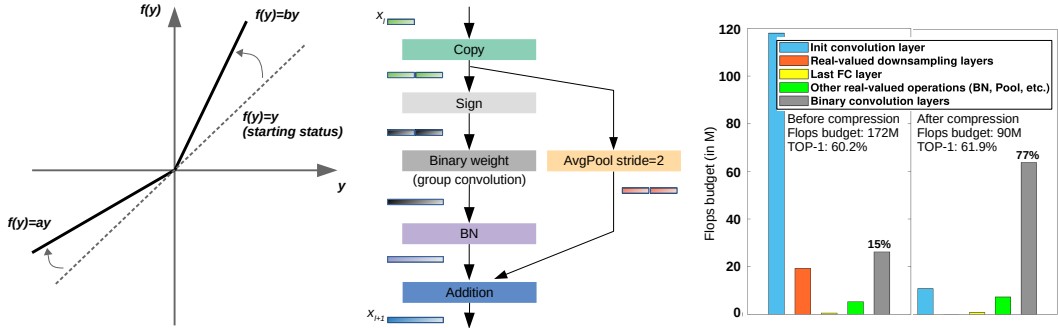

Figure 3: *Left*: The proposed FPReLU, slops in both sides are started from 1 (equal to an identity map) and adaptively learned in channel-wise. *Middle*: Reduction block with group binary convolution and cheap downsampling layer (for normal block, the original binary convolution layer in Fig. 1 is also implemented with groups in the derived models). *Right*: Computational budget distributions for original and binary-purified FTBNN. The computational budget of a original floating point multiply operation (FLOP) in a convolution is reduced by 64 times if the operation is binarized (using *XNOR-Count* instead). Based on that, when measuring the computational budget, the budget of binary operations (BOPs) are converted to equivalent of FLOPs with a factor of $1/64$. 15% and 77% denote the corresponding budget percentages for BOPs.

adding specific activation functions. Contrary to Tang et al. (2017) and Bulat et al. (2019), we find that traditional ReLU is still more effective than PReLU with a stronger non-linear property. While at the same time, immoderately inserting ReLUs in the network can also cause divergence and deterioration, as the information collapse phenomenon (Sandler et al., 2018) become serious. To alleviate that, we extend PReLU to *Fully Parametric* ReLU (FPReLU) where the slops for both negative and positive activations are learnable (Fig. 3), to stably maximize BNN's flexibility.

## 2.2 TOWARDS A MORE COMPACT BNN WITH BINARY PURIFICATION

Real-valued operations in BNN are often needed to drive and preserve the information flow (*e.g.*, binarization for the 1x1 convolutional shortcuts in the reduction block of Bi-Real Net will interrupt the continuous full-precision information flow, the damage of which is hard to recover in the following layers (Bethge et al., 2019)).

From Fig. 3, we find that the most computation-consuming parts among the real-valued operations in FTBNN are the initial convolutional layer and the real-valued 1x1 convolutional shortcuts. Thereby we make the following two changes to relieve the computation burden for the baseline.

**Initial Conv:** Inspired by Bethge et al. (2020), in the full-precision initial convolutional layer, we set the kernel size to 3x3 and constrain the number of output channels (only half of that in the baseline in our experiments). To facilitate building BNNs with different width configurations, as we will discuss later, we add an extra binary convolutional layer after the initial convolutional layer as a bridge that can arbitrarily expand the initial width (see appendix for more detail).

**Downsampling shortcuts:** Similar to Liu et al. (2020b), we propose a copy-and-paste strategy to avoid a convolutional downsampling shortcut to save computation while preserving the information flow. In a reduction block where the resolutions of feature maps are halved and the number of filters is doubled, we firstly concatenate the duplicated input volume to extend the input, then employ a 3x3 average pooling with stride=2 to construct the shortcut (Fig. 3).

## 2.3 FEATURE AGGREGATION WITH GROUP EXECUTION

As discussed by Phan et al. (2020), binary operation restricts BNN's feature representation only using limited discrete values. Simply enlarging the network by increasing the number of channels can be very expensive since a $N\times$ widening leads to a $N^2\times$ growth in computational budget. We exploit the group convolution (Xie et al., 2017) to enable a wider network (Fig. 3), which we believe

can strengthen BNN's representational ability, while keeping the total overhead unchanged. The differences to other similar works are as follows.

**Compared with methods using multiple bases or branches:** Lin et al. (2017) and Zhuang et al. (2019) gathered results from multiple branches or bases to enrich the feature representation for binary convolutions. Their methods can be equally viewed as running several independent networks parallelly followed by a summation, while ours only uses a single pipeline without introducing additional computational overhead. In the form of information aggregation, concatenation is used in group convolution, which is different to summation adopted in these methods.

**Compared with methods leveraging group convolutions:** Group convolution is also incorporated in recent approaches to facilitate binary network architecture search (NAS) (Phan et al., 2020; Bulat et al., 2020), while unfortunately a proper ablation study is missing, making the benefits from the group convolution unclear under the whole framework where the resulted advantage may be biased towards the NAS technology itself. Our study can be regarded as such an specific investigation to make a better understanding of group convolution when designing BNNs.

## 3 EXPERIMENTS AND DISCUSSES

**FTBNN Baseline:** As shown in Fig. 1, to give the model enough non-linearity while maintaining training stability, we insert a ReLU after every four blocks and for other blocks, FPReLU is adopted.

**Compact and efficient derived models:** Based on our baseline and the proposed binary purification pipeline, we set the number of output channels in the initial convolutional layer as 32. A bridge binary convolutional layer is then connected, followed by the blocks with group convolution as shown in Fig. 3 (also see the appendix) and the non-linear modules. Finally, a FC layer followed by a soft-max layer is applied to do the classification. In order to observe the effect of group convolution, we elaborate different computational budgets for BNNs (calculated in FLOPs as stated in Fig. 3, following Liu et al. (2018) and Rastegari et al. (2016)) by varying width and depth.

**Replacing *XNOR-Count* when ReLU is used:** In Fig. 1 (b), activation values before Sign are either 0 or positive if ReLU is applied at the end of the previous block. Then, all the 0s would be assigned to the lower binary value with the Sign function, eliminating the effect of ReLU in the residual branch. In this case, we have designed particular bit operations, which is illustrated in the appendix.

All the models are simply trained using Adam optimizer (Kingma & Ba, 2014) for 60 epochs from scratch with a starting learning rate 0.01, which is decayed in a multi-step way (at epoch 45 and 55, with the decaying rate 0.1) for the baseline, and a cosine shape scheme (towards 0 at the end of training) for the derived models. However we note there is no big difference between this two ways. The weight decay is set to 0 for all the binary convolutional layers. We use STE (Hubara et al., 2016) for backward propagation with value clipping in range (-1.2, 1.2) for both weights and activations. The Pytorch library (Paszke et al., 2019) is applied for implementation. Finally, following Bi-Real Net, we use the large-scale challenging ImageNet benchmark (ILSVRC 2012, Deng et al. (2009)) in our experiments. A more detailed description for the implementation can be found in the appendix.

### 3.1 ABLATION ON FTBNN BASELINE

The most similar one of the proposed baseline is the Bi-Real Net structure. Where, the authors used a 2-stage training scheme to get a good initialization for BNN, a magnitude-aware gradient approximation was also considered during weights updating. Finally, the trained scaling factors are absorbed into the BN layers. The whole training procedure was carefully tuned in order to fully exploit the model's capacity. It can be seen in Table 3 that when applying our naive training scheme, the Bi-Real Net structure (Fig. 1 (a)) deleteriously degraded (from 56.4% to 52.3% on Top-1 accuracy). The main deficiency of this structure is the lack of a proper non-linearity introduction, which is demonstrated to be indispensable for BNNs. In addition, our structure without any non-linear module performs slightly better than the Bi-Real Net structure (52.5% *vs.* 52.3%), demonstrating the benefits of a weaker scaling mechanism from the BN layers.

The following experiments then focus on the effectiveness of different non-linear modules. From Table 3, we can find that the ReLU function is still preferred with a stronger non-linear property, that is able to bring in more discriminative power. For example, only inserting 4 ReLUs can outperforms

Table 2: Binary convolutions in FTBNN baseline based on ResNet-18 structure. BConv$x$-$y$ indicates $y$th block in the $x$th stage, where the size and number of filters are also given. The learned coefficients of FPReLU after each block are listed in the form of *mean [minimum maximum].*

| Layer | | Learned coefficients | | Layer | | Learned coefficients | |
|---|---|---|---|---|---|---|---|
| | | Negative | Positive | | | Negative | Positive |
| BConv1-1 | 3×3, 64 | 1.82 [0.48 4.63] | 1.16 [0.63 2.19] | BConv3-1 | 3×3, 256 | 0.84 [0.47 2.07] | 0.95 [0.61 1.84] |
| BConv1-2 | 3×3, 64 | 0.95 [0.00 2.69] | 0.72 [0.34 1.72] | BConv3-2 | 3×3, 256 | 0.86 [0.44 2.85] | 0.79 [0.46 1.51] |
| BConv1-3 | 3×3, 64 | 0.87 [0.33 4.49] | 0.75 [0.26 1.40] | BConv3-3 | 3×3, 256 | 0.76 [0.04 1.97] | 0.78 [0.28 2.50] |
| BConv1-4 | 3×3, 64 | ReLU | | BConv3-4 | 3×3, 256 | ReLU | |
| BConv2-1 | 3×3, 128 | 0.91 [0.54 2.80] | 0.93 [0.56 2.04] | BConv4-1 | 3×3, 512 | 0.96 [0.54 2.86] | 0.90 [0.14 2.05] |
| BConv2-2 | 3×3, 128 | 0.93 [0.45 3.16] | 0.83 [0.54 1.70] | BConv4-2 | 3×3, 512 | 1.04 [0.44 4.66] | 0.67 [0.00 1.44] |
| BConv2-3 | 3×3, 128 | 0.81 [0.00 1.82] | 0.86 [0.46 1.69] | BConv4-3 | 3×3, 512 | 0.68 [0.23 2.80] | 0.95 [0.23 3.51] |
| BConv2-4 | 3×3, 128 | ReLU | | BConv4-4 | 3×3, 512 | ReLU | |

Table 3: Ablation study for FTBNN baseline based on the ResNet 18 architecture.

| Model | Top-1 (%) | Top-5 (%) |
|---|---|---|
| Bi-Real Net reported by Liu et al. (2018) | 56.4 | 79.5 |
| Bi-Real Net with our naive training scheme | 52.3 | 75.9 |
| No explicit non-linear modules | 52.5 | 76.0 |
| All configured with PReLU | 58.9 | 81.1 |
| All configured with FPReLU | 59.3 | 81.3 |
| All configured with ReLU (lr=0.00015, diverge if lr is too big) | 45.9 | 70.5 |
| Partially configured with ReLU | 59.6 | 81.4 |
| Partially configured with ReLU & PReLU in others | 59.7 | 81.6 |
| Baseline (Partially configured with ReLU & FPReLU in others) | **60.2** | **82.0** |
| Full-precision of ResNet-18 (our implementation) | 69.8 | 89.2 |

the one with all blocks connected with PReLU by 0.7% in Top-1 accuracy. Note that ReLU is a special case of PReLU, indicating ReLU is already a decent local optima for PReLU by giving a strong non-linearity, making the optimization easier. At the same time, if the usage of ReLU is limitless, the BNN would also be prone to divergence, showing that unlike the full-precision counterparts, binarized architectures are essentially vulnerable, an abuse of ReLU which causes severe information collapse in the feature space should be avoided.

A straightforward but effective way to control the non-linearity, so that model's discriminative power can be maximized, and the stability is also guaranteed, is to control the number of ReLUs and additionally introduce weaker non-linear functions. Here, we show that compared with PReLU, the proposed FPReLU is an ideal solution to meet this requirement. As shown in Fig. 3, FPReLU starts from an identity function and gradually evolves during training to compensate the non-linearity in both sides of activations. Table 2 lists the ultimately learnt coefficients of the FPReLU functions. These values for both side of the functions vary around 1 with a fairly wide range, some even approach to 0 or a big value (*i.e.*, 5), meaning that FPReLU is competent to tune the network into a balanced point automatically, since both slops for negative and positive activations are considered.

### 3.2 THE DERIVED MODELS

In this section, we evaluate the effectiveness of the derived models with binary purification by adjusting the number of groups. Table 4 *left* shows the influences of group convlutions on binary-purified BNNs with varying settings of depth and group numbers, by constraining the computation costs similar. It can be found that the simplest derived models without using group convolutions (*i.e.,*, G = 1) can surpass the proposed FTBNN baseline by a large margin (a 1.7% improvement on Top-1 accuracy with computational budget almost halved based on ResNet 18), which strongly demonstrates the effectivenss of the proposed binary purification strategy. It also shows that there is no influence of the group execution for relatively shallow binary networks under different budget constraints. As going deeper, the group convolution begin to diverge the corresponding models with notable characteristics. On one hand, when observing the training accuracy, increasing the number of groups can obviously introduce a better fitting, indicating an increasing representational capacity of the BNN. The phenomenon can be found where the layer $\geq 34$ regardless of the model complexity.

Table 4: *Left*: Experimental results of the derived structures with different group and budget configurations, with Top-1 accuracy (in %) recorded, where the widths of networks are adjusted to meet the budget requirements. For a clearer illustration, we also give the number of parameters for each model: Params = number of floating point paramters + number of binary parameters / 32. *Right*: Comparison with state-of-the-arts. (W/A) means how many bits used in the weights (W) and activations (A), (1/1) $\times n$ indicates $n$ branches or bases or individual models were adopted to expand the network capacity for BNN. Our method with $*$ indicates knowledge distillation and multi-step training were used (refer to appendix for detail).

| Budget (in M) | Layer | Group | FLOPs (in M) | BOPs (in M) | Params (in M) | Top-1 (Train) | Top-1 (Val) |
|---|---|---|---|---|---|---|---|
| ~90 | 18 | 1 | 19.0 | 4516 | 1.66 | 61.0 | 61.9 |
| | 18 | 3 | 23.1 | 4580 | 2.06 | 61.6 | 61.9 |
| | 18 | 5 | 25.8 | 4301 | 2.28 | 62.2 | 61.9 |
| | 18 | 8 | 28.7 | 3988 | 2.55 | 61.5 | 61.1 |
| | 34 | 1 | 19.3 | 4176 | 1.32 | 58.8 | 61.0 |
| | 34 | 3 | 23.8 | 4005 | 1.59 | 60.6 | 62.0 |
| | 34 | 5 | 27.0 | 3829 | 1.79 | 61.8 | 62.5 |
| | 34 | 8 | 30.1 | 3577 | 1.97 | 61.9 | 62.0 |
| ~135 | 18 | 1 | 21.1 | 7399 | 2.42 | 66.1 | 64.8 |
| | 18 | 3 | 26.2 | 7285 | 2.88 | 66.6 | 64.5 |
| | 18 | 5 | 29.9 | 7031 | 3.22 | 67.2 | 64.2 |
| | 18 | 8 | 34.0 | 6717 | 3.60 | 66.4 | 63.4 |
| | 34 | 1 | 21.6 | 6994 | 1.99 | 63.5 | 64.5 |
| | 34 | 3 | 27.4 | 6639 | 2.31 | 65.8 | 65.3 |
| | 34 | 5 | 31.5 | 6412 | 2.56 | 66.8 | **65.5** |
| | 34 | 8 | 36.0 | 6090 | 2.83 | 67.3 | 65.0 |
| | 44 | 1 | 22.1 | 6847 | 2.07 | 62.4 | 63.3 |
| | 44 | 3 | 28.5 | 6512 | 2.33 | 65.3 | 64.8 |
| | 44 | 5 | 32.9 | 6251 | 2.55 | 66.4 | 65.3 |
| | 44 | 8 | 37.5 | 5947 | 2.76 | 66.9 | 65.3 |

| Method | Bitwidth (W/A) | Budget (in M) | Top-1 (in %) |
|---|---|---|---|
| TTQ (Zhu et al., 2017) | 2/32 | | 66.6 |
| BWN (Rastegari et al., 2016) | 1/32 | | 60.8 |
| RQ ST (Louizos et al., 2019) | 4/4 | | 62.5 |
| ALQ (Qu et al., 2020) | 2/2 | N/A | 66.4 |
| SYQ (Faraone et al., 2018) | 1/8 | | 62.9 |
| HWGQ (Cai et al., 2017) | 1/2 | | 59.6 |
| LQ-Net (Zhang et al., 2018) | 1/2 | | 62.6 |
| ABC-Net (Lin et al., 2017) | (1/1)×5 | | 65.0 |
| CBCN (Liu et al., 2019) | (1/1)×4 | N/A | 61.4 |
| Group-Net (Zhuang et al., 2019) | (1/1)×4 | | 66.3 |
| BNN Ensemble (Zhu et al., 2019) | (1/1)×6 | | 61.0 |
| XNOR-Net (Rastegari et al., 2016) | 1/1 | | 51.2 |
| Bi-Real Net-18 (Liu et al., 2018) | 1/1 | | 56.4 |
| XNOR-Net++ | 1/1 | | 57.1 |
| CI-BCNN (Wang et al., 2019) | 1/1 | ~163 | 59.9 |
| BONN (Gu et al., 2019) | 1/1 | | 59.3 |
| IR-Net (Qin et al., 2020) | 1/1 | | 58.1 |
| **FTBNN Baseline (ours)** | **1/1** | | **60.2** |
| **FTBNN Baseline $*$ (ours)** | **1/1** | | **63.2** |
| Bi-Real Net-34 (Liu et al., 2018) | 1/1 | 193 | 62.2 |
| Binary MobileNet (Phan et al., 2020) | 1/1 | 154 | 60.9 |
| Real-to-Bin (Martinez et al., 2020) | 1/1 | 183 | 65.4 |
| **FTBNN derived (ours)** | **1/1** | **132** | **65.5** |

On the other hand, the positive effect for deeper networks can also activate a better accuracy on the validation set, meaning that an increase in discriminative power are also accompanied. All the experiments on deeper BNNs with Group = 5 show a clear accuracy boost than those without group execution (Group = 1). *E.g.*, from 61.0% to 62.5% for Layer = 34, and from 63.3% to 65.3% for Layer = 44.

However, an increasing representational capacity may also lead to a higher risk of dropping in generability. For some cases, the accuracy on validation set is higher than that reported on the training set, due to the regularization nature of binary operations. With group execution, such effect of binarization is gradually weakened, leading to overfitting issues.

### 3.3 COMPARISON WITH STATE-OF-THE-ARTS

Here we compare the proposed FTBNN with existing state-of-the-art methods as illustrated in the right side of Table 4, including the low-bit quantization methods, multiple binarizations methods, other binary frameworks based on the ResNet-18 architecture, *etc*. Interestingly, we find that some of our derived models are able to achieve superior or comparable performance when compared with the methods using multiple bases or branches, while with far less computational overhead, again showing the effectiveness and efficiency of the proposed method.

## 4 PERSPECTIVES

Here, we raise some possible insights from this work: a) To further enhance the model's discriminative ability through non-linearity, we wonder is it the best way to combine ReLU and FPReLU? Is it affected by other elements, such as depth, width and block orders? b) To what extent can we do the binary purification? As we still have much proportion of real-valued operations in the derived models. c) As shown in our experiments on BNN, group convolution accompanies with overfitting. How to alleviate overfitting for BNNs should also be considered in future works.

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

## A    MORE STRUCTURE INFORMATION

Fig. 4 gives more structure information of FTBNN and the derived models.

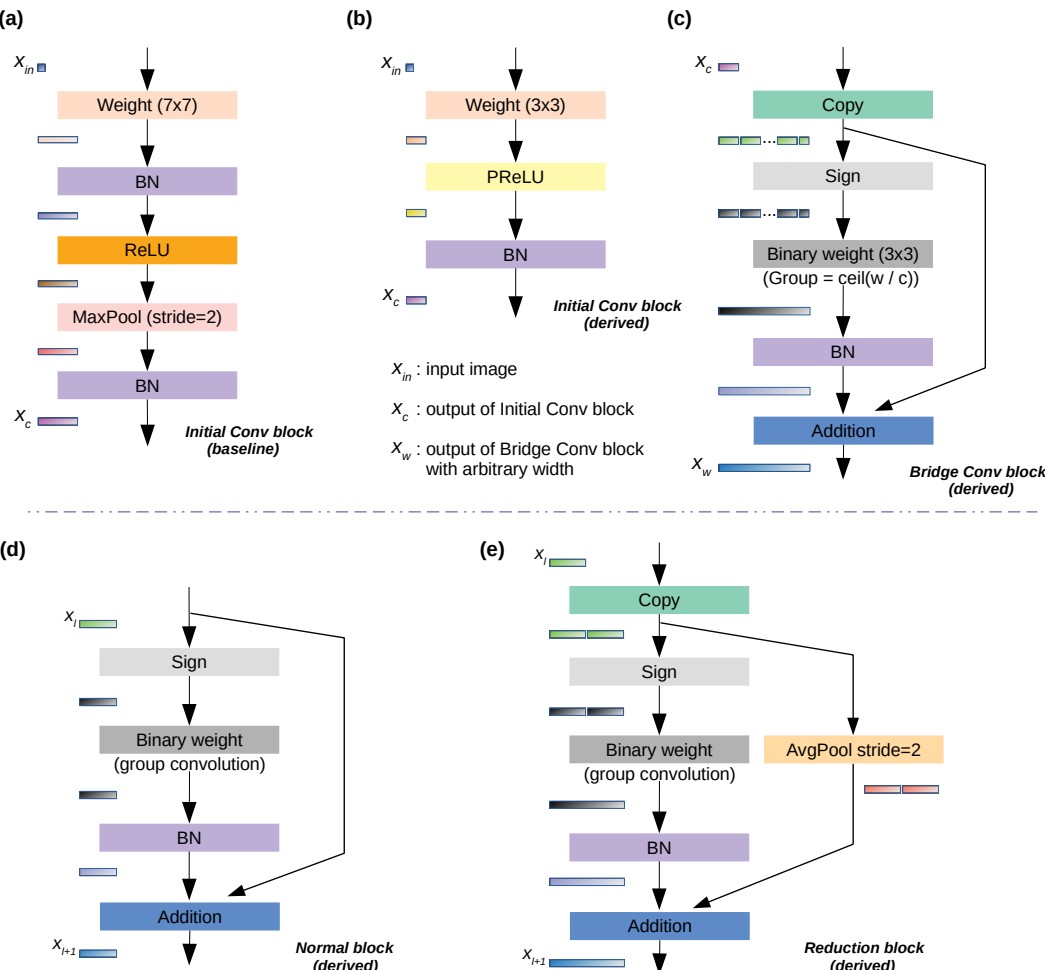

Figure 4: (a): Initial convolutional block for FTBNN baseline with a 7x7 convolutional layer, the output of which has 64 channels. (b): Initial convolutional block for the derived models with a 3x3 convolutional layer, the number of channels in the output of which is halved compared with the baseline. (c): Bridge convolutional block for the derived models, the number of channels in the output of which can be arbitrary. c and w represent the channel numbers. (d): Normal block for the derived models. (e): Reduction block for the derived models (in our experiments, we set the actual group number to 2 in the reduction block for settings *Group = 1* and *Group = 2* in Section 3.2).

## B    REPLACING *XNOR-Count* OPERATIONS IN FTBNN AS NEEDED

Suppose we are using the following Sign function to do binarization when doploying our models:

$$v_b = \text{Sign}(v_r) = \begin{cases} T, & v_r > 0 \\ F, & v_r \leq 0 \end{cases}, \tag{2}$$

where $v_r$ and $v_b$ represents the real-valued and binarized values (can be either of weights or activations) respectively. As we mentioned in Section 3, the input of a basic block will be truncated such that the values of which are either 0 or positive, if a ReLU is connected after the previous block.

Supposing the binarized weights in the current block is $w_b = [T, F, T, F, T]$ and the input activations before truncation is $x_r = [-3, -2, 1.5, 2, -1.2]$, and truncated to $x_r = [0, 0, 1.5, 2, 0]$ after ReLU. According to the Sign function, the activations will then be binarized to $x_b = [F, F, T, T, F]$.

If *XNOR-Count* operation is used, the result of binary convolution is (the *popcount* function counts the bits of $T$):

$$
\begin{aligned}
x_o &= \text{popcount}(x_b \text{ xnor } w_b) \\
&= \text{popcount}([F, F, T, T, F] \text{ xnor } [T, F, T, F, T])) \\
&= \text{popcount}([F, T, T, F, F]) \\
&= 2.
\end{aligned}
\tag{3}
$$

The result is the same as if the ReLU is not used, because without ReLU, the original input activations will still be binarized to $x_b = [F, F, T, T, F]$.

Actually, our goal is to make the model trained in PyTorch be deployed using bit operations. Thus, the output of convolution in PyTorch should be the same as when using bit operations. Noting the definition of *torch.sign()* in Appendix E is:

$$
torch.sign(x) = \begin{cases} -1 & \text{if } x < 0, \\ 0 & \text{if } x = 0, \\ 1 & \text{if } x > 0. \end{cases}
\tag{4}
$$

In the above case, the process of convolution in PyTorch is:

Step 1, based on Eq. 4 and Appendix E, with the presence of ReLU, $x_r$ will be binarized to: $x_b^p = [0, 0, 1, 1, 0]$;

Step 2, convolution between $x_b^p$ and $w_b^p$ ($*$ is the convolution function):

$$
\begin{aligned}
w_b^p &= [1, -1, 1, -1, 1] \\
x_o^p &= x_b^p * w_b^p = 0.
\end{aligned}
\tag{5}
$$

Obviously, the output from Eq. 3 does not match the output $x_o^p$. To match $x_o^p$ when deploying with bit operations, we can use the following calculation:

$$
x_o = \text{popcount}(x_b \text{ and } w_b) - \text{popcount}(x_b \text{ and } !w_b).
\tag{6}
$$

Therefore, the output of bit operations is:

$$
\begin{aligned}
x_o &= \text{popcount}([F, F, T, T, F] \text{ and } [T, F, T, F, T]) - \text{popcount}([F, F, T, T, F] \text{ and } [F, T, F, T, F]) \\
&= \text{popcount}([F, F, T, F, F]) - \text{popcount}([F, F, F, T, F]) \\
&= 0 = x_o^p.
\end{aligned}
\tag{7}
$$

In addition, Eq. 6 may double the number of BOPs than the original binary convolution which only uses *XNOR-Count*. We have considered that when calculating the computational budgets for our models using doubled BOPs when ReLU is used.

## C    EXPERIMENTAL CONFIGURATION FOR TRAINING

**Data augmentation:** The large-scale ImageNet dataset (ILSVRC 2012, Deng et al. (2009)) is used in our experiments as the benchmark, consisting of 1.28 million images for training and 50,000 images for validation. We adopt the standard data augmentation scheme as in He et al. (2016) during training, including random scale augmentation, random crop to 224×224, random horizontal flip with probability 0.5 and per-pixel mean value subtraction. During testing, we resize the image such that the shorter side is 256 while keeping the original aspect ratio, crop a 224×224 patch in the center, and apply the same per-pixel value subtraction as the training set.

**Multi-step training:** This training scheme is only used to enhance the proposed FTBNN baseline, demonstrating that the baseline is essentially orthogonal to other training insights for BNNs (for other cases in our experiments, the naive training scheme with 60 epochs mentioned in Section 3

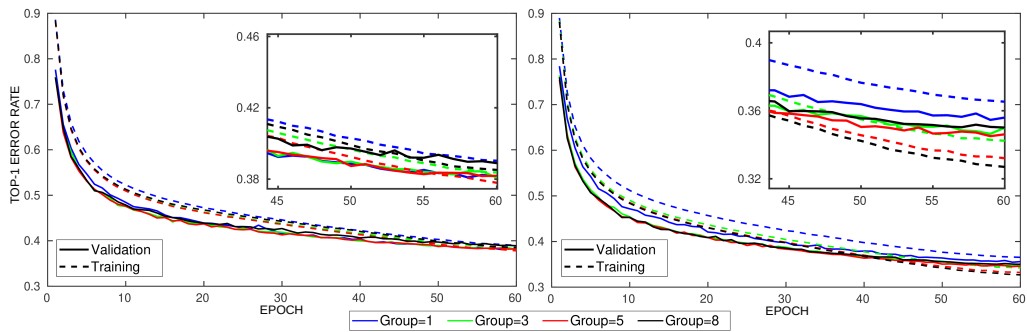

Figure 5: *Left:* Training curves for Layer = 18 with computational budget ∼90M. *Right:* Training curves for Layer = 34 with computational budget ∼135M. Group execution has a positive effect for deeper BNN structures as discussed in Section 3.2.

is applied). Following Martinez et al. (2020), we adopt the following teacher-student knowledge distillation with attention mapping for multi-step training. Specifically, we firstly train a full-precision model by keeping the weights real-valued and use Tanh function instead of Sign function for activations. Then, a two-stage training scheme is used. In stage 1, we adopt Sign function for activations, while keeping the weights real-valued. In stage 2, both weights and activations are binarized with the Sign function. In both stages, we use the output of the original ResNet-18 model as the soft labels when calculating cross-entropy loss.

## D   TRAINING CURVES IN GROUP EXECUTION

The training curves for FTBNN with group execution are shown in Fig. 5.

## E   CORE CODE USED IN OUR IMPLEMENTATION WITH PYTORCH

```python
import torch
import torch.nn as nn
import torch.nn.functional as F

######   Binarization for activations   ######

class Sign(nn.Module):

    def __init__(self, bound=1.2):
        super(Sign, self).__init__()
        self.bound = bound

    def forward(self, x):
        out = torch.clamp(x, -self.bound, self.bound)
        out_forward = torch.sign(x)
        y = out_forward.detach() + out - out.detach()
        return y

######   Binarization for weights and binary convolution   ######

class BConv(nn.Module):

    def __init__(self, in_chn, out_chn, kernel_size=3, stride=1,
            padding=1, groups=1, bound=1.2):
        super(BConv, self).__init__()
        self.stride = stride
        self.padding = padding
```

```python
        self.groups = groups
        self.bound = bound

        shape = (out_chn, in_chn // groups, kernel_size, kernel_size)
        self.weights = nn.Parameter(torch.Tensor(*shape))
        torch.nn.init.xavier_normal_(self.weights, gain=2.0)

    def forward(self, x):
        clipped_weights = torch.clamp(self.weights, \
                -self.bound, self.bound)
        binary_weights_no_grad = torch.sign(self.weights)
        binary_weights = binary_weights_no_grad.detach() + \
                clipped_weights - clipped_weights.detach()

        out = F.conv2d(x, binary_weights, stride=self.stride,
                padding=self.padding, groups=self.groups)
        return out

###### FPReLU ######

class FPReLU(nn.Module):

    def __init__(self, chn):
        super(FPReLU, self).__init__()
        self.l_weights = nn.Parameter(torch.ones(1, chn, 1, 1))
        self.r_weights = nn.Parameter(torch.ones(1, chn, 1, 1))

    def forward(self, x):

        x = torch.where(x > 0, x * self.r_weights, x * self.l_weights)
        return x
```

