# OpenReview forum: "FTBNN: Rethinking Non-linearity for 1-bit CNNs and Going Beyond"
_ICLR.cc/2021/Conference — Reject_

### Official Review · AnonReviewer2 · 2020-10-16
**Official Blind Review #2**

**Rating:** 4
**Confidence:** 5

**Review:**

---paper summary---:

This paper proposes to improve the BNN’s discriminative ability by introducing additional non-linearities. In addition, the paper exploits the group convolution to enable a wider network, which can strengthen BNN’s representational ability, while keeping the total overhead unchanged.


---Pros---:

This paper introduces some practical methods to improve the performance of BNNs. In particular, the additional FPReLU is convincing. Moreover, the paper shows that grouped convolutions can be applied to wider BNNs to increase the representational capability while keeping the same complexity.

---Cons---:

1:  This paper is incremental with limited new technical insights.

a) Adding the nonlinearity can improve the representational capability of BNNs has been extensively explored in the literature. Specifically, ReActNet [Liu et al. 2020b] inserts additional RPReLU after each binary convolution to increase the non-linearity;  [A1; Martinez et al., 2020; Tang et al. 2017]  add PReLU as the nonlinearity;  Group-Net [Zhuang el al. 2019] and XNOR-Net [Rastegari et al., 2016] argue that adding additional ReLU is important to BNNs performance.
b) Varying width and/or depth has been studied in previous fixed-point quantization/BNNs literature, especially in NAS-based methods [Bulat et al. 2020;  A2]. The original idea comes from EfficientNet.
c)  Some other tricks such as replacing the 1x1 downsampling shortcut with pooling have been widely used in the community.

2:   Some arguments need theoretical proofs. For example, the authors argue that “despite the big quantization error made by quantization, the binary model can achieve much higher accuracy than the real-valued model with no quantization error”. In other words, minimizing the quantization error can hinder the discriminative ability of BNNs, which is the main point of this paper. This observation is interesting, but needs further theorems to further explore whether the quantization error has some relations with the predicted accuracy under some constraints. If zero quantization error cannot lead to a good result, then what should be the best trade-off? I encourage the authors to further explore this issue. At the current stage, it is far from enough.

3: The experimental results in Table 4 may have mistakes. The paper claims that BOPs are converted to equivalent FLOPs with a factor of $1/64$. However, why do smaller BOPs correspond to larger FLOPs?

4:  The necessary “AND-count” operations may have technical issues. The AND-Count with values binarized to {0,1} should be equivalent to XNOR-Count with the values binarized to {-1, 1}, with a scalar difference. The authors can derive by themselves to verify this. However, the formulations in Eq. (3) and Eq. (4) are not equivalent if both values are binarized to {-1, 1}.

5: More experimental results on deeper networks (e.g., ResNet-50, -101) on ImageNet are needed to justify the effectiveness of the method.  In addition, the comparisons with RELU [Zhuang el al. 2019; Rastegari et al., 2016], PReLU [A1; Martinez et al., 2020; Tang et al. 2017],  RPReLU [Liu et al. 2020b] (optional) should be included.

6: Some typos. For example, “inheriting exiting advanced structures” → “inheriting existing advanced structures”; “Base on this consideration” → “Based on this consideration”.

References:

[A1]: Bayesian Optimized 1-Bit CNNs, in ICCV2019

[A2]: Joint Neural Architecture Search and Quantization, arXiv:1811.09426v1

---

> ### Author Response · Authors · 2020-11-14
> **Thanks for your detailed feedbacks**
>
> We greatly appreciate your review and valuable comments, please see our replies as follows:
>
> Q1: Compare with other methods which also explored the non-linearity for BNNs.
>
> R1: For comparison with ReActNet, we kindly request you to refer to R3 for Review#3. As you mentioned, PReLU was widely adopted in many previous works to improve the predictive performances for BNN structures. Despite their successes, those works are more like to use PReLU as an insight while may with lack of further explorations. This paper indeed compared the PReLU and ReLU based BNN structures (please also refer to R2 for Review#3 and Section 3.1) and find more about how to fully exploit the benefit of non-linearity.
>
> Q2: Varying width/depth has been studied in previous fixed-point quantization/BNNs literature, especially in NAS-based methods.
>
> R2: NAS-based methods are more about how to find the optimal depth/width configurations with a certain computational recourse constraint, to get the highest accuracy. And they may not use group convolutions. While our work is more about how group convolution can affect the performance of BNN with a certain computational cost constraint. In addition, in order to give a more detailed analysis, we adopt different depths in BNNs.
>
> Q3: Some other tricks such as using the pooling downsampling layers have been widely used.
>
> R3: Yes, we actually used the pooling downsampling shortcuts only as a common tool to make the BNN more compact. We are more focusing on whether it causes accuracy drop or not while still maintaining stability and efficiency for our baseline and how far the baseline can go further.
>
> Q4: To further explore whether the quantization error has some relations with the predicted accuracy under some constraints and what should be the best trade-off to control the quantization error.
>
> R4: To be frank, it is non-trivial to provide the theoretical proofs to explain how the quantization error affect BNNs. An intuitive interpretation on why we should not minimizing it to 0 is that in this case, the whole BNN, which has no explicit non-linear modules, will become a linear function. On the other hand, we also avoid taking no constraints on the quantization error. What we did is to weaken the scaling mechanism, by removing the explicit scaling factors and only using BN to learn how to scale. This makes some sense as we found that we can achieve slightly better performance in that way (52.5% vs. 52.3% on ImageNet). While this was still not optimal. The further introduced explicit non-linear modules can be regarded as a compensation to deliver a better trade-off.
>
> Q5: Why do smaller BOPs correspond to larger FLOPs?
>
> R5: We think there is a misunderstanding in that. The calculation of FLOPs for a model is: Total FLOPs = FLOPs in floating point operations + BOPs in binary operations / 64. The calculation is following [1, 2].
>
> Q6: The necessary “AND-count” operations may have technical issues.
>
> R6: We have made some modifications to deal with the case where ReLU is used, please refer to R1 for Reviewer#4 and the modified appendix. Our basic idea is to make the model trained with PyTorch able to be deployed in real devices with bit operations. I have some difficulties in understanding “The AND-Count with values binarized to {0,1} should be equivalent to XNOR-Count with the values binarized to {-1, 1}” since we think 0 or -1 will not affect the actual binary operations because they have the same meaning when conducting bit operations. Could you explain that more?
>
> Q7: Some typos.
>
> R7: Thanks for pointing that out. We have corrected these typos in the revised version.
>
> [1] Rastegari et al. Xnor-net: Imagenet classification using binary convolutional neural networks, ECCV 2016.\
> [2] Liu et al. Bi-real net: Enhancing the performance of 1-bit cnns with improved representational capability and advanced training algorithm, ECCV 2018.

---

### Official Review · AnonReviewer4 · 2020-10-28
**Nice work, but more analysis is required for acceptance**

**Rating:** 5
**Confidence:** 4

**Review:**

The author proposed an improved binary neural network (BNN) model.  FPReLU which increases the discriminative ability of BNN and group convolution make the proposed BNN architecture achieve better accuracy than prior works under similar operation budget condition. Other techniques like knowledge distillation can further increase the model performance.

4-5 % top-1 accuracy improvement compared to prior models is an impressive result. I think reducing the burden of floating-point operations by editing first convolution layer and downsampling shortcuts is another good point, because the computation of uncompressed first convolution layer is easy to overlooked, but can be the bottleneck in the case of end-to-end inference.

The paper was clearly well-written, but I still have a concern or questions about the parts described below. I hope the authors to solve my concerns and questions so that my evaluation would be changed.

1. According to the sign function in Appendix E, its output value is either 0 or 1 if its input is from ReLU and is either -1 or 1 if its input is from PReLU, FPReLU or not passed through a non-linear function. According to [1],  better performance can be achieved by using [0,1] activation in BNN, so my major concern is that the accuracy improvement of Bi-Real Net V2 compared to Bi-Real Net may come from the fact that both [0,1] and [-1,1] activation are used in a network.  Note that 'Baseline' and 'Partially configured~' models in Table 3 use both [-1,1] and [0,1] activation.
2. (-1.2,1.2) clipping can seem to be unfamiliar, because usually (-1.0, 1.0) range is used for BNN. I have the question how that range was decided.
3. One of the factor which can decide the model accuracy is the number (or total bits) of parameters, so I think the performance comparison among BNN models will be more clear if the authors add that information of each BNN models.

Minor comment:

In Appendix C, it is stated that original ResNet-18 model is used as a teacher model in all stages, which is different from [2] where the model trained at the previous stage is used as a teacher. This is the difference between using a teacher which has a good classification ability and using a teacher which resembles the student. I question if the authors had any reason to choose the former method and how two training cases affect to BNN performance.

Reference

[1] Peisong Wang, et al. Sparsity-Inducing Binarized Neural Networks. AAAI, 2020.
[2] Brais Martinez, et al. Training binary neural networks with real-to-binary convolutions. ICLR, 2020.

---

> ### Author Response · Authors · 2020-11-14
> **We have changed the bit operations when ReLU is used and added the model size information in the revised version**
>
> We really appreciate your review and detailed feedbacks. Please see the replies of your concerns as follows:
>
> Q1: The accuracy improvement may come from the fact that both [0, 1] and [-1, 1] activations are used in a network.
>
> R1: Thanks for pointing out this important question. Because in the [0, 1] activations, there is no -1, we can then avoid using ternary activations when deploying our model to binary operation supported devices, by using particular bit operations. So the trained model is still a BNN. Please consider the following case (also hold for other cases), the output of convolution in PyTorch should be the same as when using bit operations (we also have modified the solution in the appendix):
>
> Activations before ReLU: -3, -2, 1.5, 2, -1.2\
> Weights: 1, -1, 1, -1, 1
>
> PyTorch:\
> Activations after ReLU + torch.sign(): 0, 0, 1, 1, 0\
> output of convolution in PyTorch: 0
>
> To match the output of convolution in PyTorch when we deploy our model, we can use the following bit operations with an additional subtraction (instead of using the number, we use T and F to represent True and False):
>
> output = popcount(A and W) – popcount(A and !W), \
> where W becomes: T, F, T, F, T\
> A becomes: F, F, T, T, F\
> The output will be popcount(F, F, T, F, F) – popcount(F, F, F, T, F) = 1 – 1 = 0
>
> It was a mistake as we mentioned in our paper to simply use AND-Count, and corrected that in our revised manuscript.
>
> In addition, the above equation may double the number of BOPs than the original binary convolution which only uses popcount(A xnor W). This takes some minor extra computational cost for our structures while not affecting the conclusions obtained in the paper, e.g., the FLOPs is from 169.3M -> 172.0M for our baseline model. We have corrected those numbers in the revised paper.
>
> Q2:  How the clipping range (-1.2, 1.2) was decided?
>
> R2: We empirically set the clipping range as (-1.2, 1.2) since we found that (-1, 1) got slightly worse performances. Other similar works also took a wider clipping range, e.g., BinaryDenseNet [1] used (-1.25, 1.25) and (-1.3, 1.3) in their methods.
>
> Q3: About the number of parameters.
>
> R3: We use the equation to calculate the number of parameters: Params = number of floating point parameters + number of binary parameters / 32. It is hard to control that both computational budget and model size keeps unchanged and we focus our analysis considering the budget. However, we can also find the number of parameters (or model size) for all our models are kept very small (ranging from 1.32M to 2.76M). We have added the model size information in the left of Table 4 for a better illustration.
>
> Q4: Why we didn’t use the training stages proposed in [2] for knowledge distillation?
>
> R4: Our structures are very similar to the original ResNet-18 because we have added some ReLU functions in the intermediate layers. To make it simpler, we directly used the original ResNet-18 model as the teacher for all the stages. This is possible that we can further increase the accuracy by following exactly the training stages as in [2].
>
> [1] Bethge et al. Binarydensenet: Developing an architecture for binary neural networks, ICCVW 2019.\
> [2] Martinez et al. Training binary neural networks with real-to-binary convolutions. ICLR 2020.

---

### Official Review · AnonReviewer1 · 2020-10-28
**The benefit of extra non-linearities in BNN**

**Rating:** 6
**Confidence:** 2

**Review:**

summary:
Summary:
The authors note that the poor performance of BNN may be due to the disappearing of the amount of non-linearity in the input-output transformation. They propose a new Bi-Real architecture where they replace the scaling factor with extra non-linearities. The performance of the model is compared with the original Bi-Real Net and further possible improved version of the proposed method.

Strengths:
Analyzing the role of non-linearity is a very nice idea. Studying BNNs is a clean setup for understanding the well-known predictive power of NN, even in real-valued settings. The computational gains associated with the purification of the convolutional layers look impressive.

Weaknesses:
The comparison is between binarized but rescaled networks and binary networks without rescaling but added non-linearity. It is not clear if the good performance of the proposed architecture is due to i) the BN step making the scaling redundant or ii) the additional nonlinearities.  In some sense, noting that the inclusion of non-linear transformations can boost the predictive power of a network is not surprising. The extra computational cost associated with the introduction of non-linear activation is not fully discussed. The comparison does not include the case where non-linearities are added to the rescaled version (the one on the left of Figure 1).

Questions:
- what happens if the binarized part of the network (between the FP first and last layer) is completely removed? Often, a two-layer network may achieve good accuracy on simple tasks such as MNIST-digit recognition.
- how does the introduction of extra non-linearity affect the computational budget in the proposed method?
- is there a figure presenting the performance of the reduced budget model shown in figure 3?
- is the benefit associated with extra non-linearities be expected to extend to other architecture than Bi-Real?

---

> ### Author Response · Authors · 2020-11-14
> **Replies to Reviewer1**
>
> We really thank you for your review and good comments. The replies of your concerns are as follows:
>
> Q1: It is not clear what the good performance of the proposed architecture is due to.
>
> R1: We can reply your concern by looking at ablation study in Section 3.1. We suspect that the explicit scaling factors used in previous works may hinder the non-linearity of BNNs, thus we remove them. Instead, we only use the scaling parameters in BN layers to take the role of scaling. We can say, the proposed architecture does have a weaker scaling mechanism compared with those using both the explicit scaling factors and the BN layers. By comparing the structures with and without the explicit scaling factors, as shown in the 3rd and 4th row on Table 3, the one without explicit scaling factors (thus only has the BN layers) achieves slightly better performance (52.5% vs. 52.3%). By adding additional non-linear modules, we can further increase the accuracy of the proposed structure. To our best knowledge, the BN layers were used in almost all the previous BNN methods.
>
> Q2: The extra computational cost of the introduced non-linear functions.
>
> R2: The extra computational cost is minor. For example, in our baseline structure, the proportion of computational cost caused by ReLU and FPReLU is 2.3M / 172.0M = 1.3%.
>
> Q3: What happens if the binarized part of the network is completely removed?
>
> R3: For MNIST-digit dataset, the “Linear” model in Figure 2 is the one without any binarization. All the layers in this model are real-valued. We can see that it is worse than the ones with binarized layers, which inspires us that the binarization functions may provide some non-linearities that benefit the network on its own.
>
> Q4: What is the performance of the reduced budget model shown in figure 3?
>
> R4: We have corrected those number in Figure 3 according to R1 for Review#4. Specifically, the comparison between the baseline model and the model with budget reduced is 60.2% vs. 61.9% on Top-1 accuracy and 172M vs. 90M. We can conclude that the baseline model can go further with budget halved while accuracy increased.
>
> Q5: Can the extra non-linearities also benefit other architectures than Bi-Real?
>
> R5: We expect an answer “yes”, and we would like to do that in our future works.

---

### Official Review · AnonReviewer3 · 2020-10-28
**Official Blind Review #3**

**Rating:** 3
**Confidence:** 5

**Review:**

This paper proposes an improvement of Binary Neural Networks from the work of Bi-RealNet (Liu et al. 2018) by utilizing the linearity of modules. The method removes the necessity of scaling factors and used a novel non-linear layer named Fully Parametric ReLU (FPReLU) to increase the capability of BNNs. In addition, similar to recent works in BNNs, the paper also utilize group convolution layers to reduce the number of parameters and save the computation's cost. The experiments are tested with image classification tasks on the large-scale dataset of ImageNet. The paper is well written and straightforward.

Although the results in ImageNet are promising and the computation is less, It raises concerns about the novelty of the work. The method used alike structure in Bi-RealNet with a bit of modification and group convolution which are already used in previous works of Liu et al. ECCV 2018 and Phan et al. CVPR 2020.

The paper used novel FPReLU but missed some comparisons with a similar idea work using PReLU[1,2].

For results in ImageNet, it would be nice if we can do more comparison and analysis with the state-of-the-art recent work [3].

In conclusion, with current manuscripts, the paper is not sufficient enough to present at the conference.

[1].  Tang et al. How to Train a Compact Binary Neural Network with High Accuracy? AAAI 2017.
[2]. Phan et al. MoBiNet: A Mobile Binary Network for Image Classification, WACV 2020.
[3]. Liu et al. ReActNet: Towards Precise Binary Neural Network with Generalized Activation Functions, ECCV 2020.

---

> ### Author Response · Authors · 2020-11-14
> **Sincerely thanks for your review and constructive comments**
>
> Sincerely thanks for your review and constructive comments, your concerns are replied as follows:
>
> Q1: The method used alike structure in Bi-RealNet and group convolution which are already used in previous works.
>
> R1: The double skip connections proposed by Bi-RealNet was commonly used in the state-of-the-art BNN methods [1, 2, 3], due to its efficiency and effectiveness. We admit we do use the double connections and hope we can go further based on that considering the non-linearities.
>
> To differ from previous methods, we have modified the name of our structures as “FTBNN” in the revised version, due to fast training speed is one of the advantages of our models.
>
> Our exploration on group convolution is different from the work of Phan et al. CVPR 2020. The latter was about leveraging the effectiveness of the heterogeneous group convolution by network architecture search, where the number of groups in each layer is not uniform. While ours is about the homogeneous configuration. The focus of our exploration on group convolution is not to find the optimal structures, but to see how group convolution can affect the performances of BNNs with a certain computational budget constrain, which may be helpful for leveraging the benefit of group convolutions when designing BNNs. From this perspective, our work can be a binary version extension of the work from [4], where the authors investigated the influences of groups convolutions for floating point networks.
>
>
> Q2: Missing some comparisons with a similar idea work using PReLU [5, 6].
>
> R2: To compare with the widely used PReLU based BNNs, we have done particular experiments shown on Table 3. PReLU can indeed improve the performances of BNNs compared with those without any non-linear modules, e.g., 58.9% vs. 52.5% on Top-1 accuracy, matching the findings of previous works. We find that PReLU is not the best non-linear configuration for BNNs, and propose the one properly configuring with ReLU and FPReLU, which can boost the performance, e.g., 60.2% vs. 58.9%.
>
>
> Q3: Comparison and analysis with the state-of-the-art recent work [7].
>
> R3: The recently proposed ReActNet is truly a great work, which also investigated the non-linearities for BNN structures. We have added the citation in our revised manuscript. While, our method and ReActNet are actually doing the investigation from two different perspectives, and leading to two different forms of the non-linear functions (FPReLU vs. RPReLU). Specifically, ReActNet was aiming to adjust the distributions of the activations to capture more semantic features. Ours is more from the motivation that simply minimizing the quantization error towards 0 could be problematic, and investigating how to exploit the non-linearity for BNNs to increase their discriminative power while ensuring training stability and efficiency.
>
> We have used a simpler training scheme compared with the one used in ReActNet. The best result (i.e., 65.5% Top-1 accuracy based on ResNet 18, Table 4) of our model was obtained by only training from scratch with 60 epochs. ReActNet adopted a multi-step training with knowledge distillation. The reported result of ReActNet based on ResNet 18 was the same as ours.
>
> [1] Qin et al. Forward and backward information retention for accurate binary neural networks, CVPR 2020.\
> [2] Kim et al. Binaryduo: Reducing gradient mismatch in binary activation network by coupling binary activations. ICLR 2020.\
> [3] Martinez et al. Training binary neural networks with real-to-binary convolutions. ICLR 2020.\
> [4] Xie et al. Aggregated residual transformations for deep neural networks. CVPR 2017.\
> [5] Tang et al. How to Train a Compact Binary Neural Network with High Accuracy? AAAI 2017.\
> [6] Phan et al. MoBiNet: A Mobile Binary Network for Image Classification, WACV 2020.\
> [7] Liu et al. ReActNet: Towards Precise Binary Neural Network with Generalized Activation Functions, ECCV 2020.

---

### Decision · Program_Chairs · 2021-01-07
**Final Decision**

**Decision:**

Reject

**Comment:**

## Description

The paper proposes an improvement to binary neural networks with real-valued skip connections between pre-activations, by introducing more flexible learnable non-linearities on the real-valued connections. The parametric non-linearity is actually linear at initialization, which makes the training easier at the beginning. Due to learnable parameters it eventually adjusts to a more complex one, able to refine the accuracy. I think this idea is a good finding.

## Review Process and Decision
The reviewers initially gave low ratings to the paper, indicating that the contribution is incremental and not fully clearly presented.
There was no detailed discussion with the authors, since the author's response and the rebuttal revision came in the very end of the discussion period. In the subsequent discussion phase the reviewer board has not indicated any major changes to the initial reviews/ranking. The AC checked the paper and supports rejection.

## Details

The authors are encouraged to improve the paper carefully addressing points proposed by reviewers.

I think the argumentation of the paper should be improved. Some explanations are intuitive, but operating with fuzzy notions and may in fact be incorrect or irrelevant. The paper should be made more precise, based on verifiable arguments.

I think the following is crucial and not made clear in the paper:
The non-linearities inserted before the sign function *do not affect the result of sign*. They indeed affect only the residual connections. Furthermore, the structure of residual connections should be fully clarified to reveal that there are complete real-valued paths all the way from the input to the network to its output, made of the residual connections with their own learnable parameters (and 1x1 convolutions) and (learnable) non-linearities and an intake from binary convolutions on the way. The learnable non-linearities can in principle improve performance just because the real-valued paths can learn better.

I paste below feedback by reviewers to author's response (I believe they would agree to share it with authors but did not find a suitable way of doing it):

## Response by R1:
I acknowledge that I read and appreciated the authors' answers to my questions. I think the idea of analyzing the role of non-linearities is nice and I tend to confirm my score. But I also agree with other reviewers that, as it is, the paper has some unclear parts and would not complain if it is rejected.

## Response by R3:
Thanks for your responses to answer my questions for the paper. I agree with the results of the proposed FBTN for improved Binary Neural Networks (BNNs). However, my concern about the novelty of using group convolution modules in BNNs has not been addressed. I think the paper is not sufficient enough to publish at the conference. So, I do not change my rating of the paper.

## Response by R4:
I maintained my rating when combining other reviews and responses to them, despite of their well response. It is still questionable whether FPReLU, one of the main contributions they claimed, actually improves the performance of BNN remarkably. In particular, this is supported by the fact that the performance of BNN on ResNet-34 which the techniques in this paper were applied does not show much difference from 'Real-to-Bin' model.